# High Prevalence of Sarcopenia in Older Trauma Patients: A Pilot Study

**DOI:** 10.3390/jcm9072046

**Published:** 2020-06-29

**Authors:** Robert C. Stassen, Kostan W. Reisinger, Moaath Al-Ali, Martijn Poeze, Jan A. Ten Bosch, Taco J. Blokhuis

**Affiliations:** Department of Traumatology, Maastricht University Medical Centre+, 6229HX Maastricht, The Netherlands; robert.stassen@mumc.nl (R.C.S.); k.reisinger@zuyderland.nl (K.W.R.); moaath.alali@mumc.nl (M.A.-A.); martijn.poeze@mumc.nl (M.P.); jan.ten.bosch@mumc.nl (J.A.T.B.)

**Keywords:** sarcopenia, polytrauma, prevalence, mortality, skeletal mass index, muscle mass

## Abstract

Sarcopenia is related to adverse outcomes in various populations. However, little is known about the prevalence of sarcopenia in polytrauma patients. Identifying the number of patients at risk of adverse outcome will increase awareness to prevent further loss of muscle mass. We utilized data from a regional prospective trauma registry of all polytrauma patients presented between 2015 and 2019 at a single level-I trauma center. Subjects were screened for availability of computed tomography (CT)-abdomen and height in order to calculate skeletal mass index, which was used to estimate sarcopenia. Additional parameters regarding clinical outcome were assessed. Univariate analysis was performed to identify parameters related adverse outcome and, if identified, entered in a multivariate regression analysis. Prevalence of sarcopenia was 33.5% in the total population but was even higher in older age groups (range 60–79 years), reaching 82 % in patients over 80 years old. Sarcopenia was related to 30-day or in-hospital mortality (*p* = 0.032), as well as age (*p* < 0.0001), injury severity score (*p* = 0.026), and Charlson comorbidity index (*p* = 0.001). Log rank analysis identified sarcopenia as an independent predictor of 30-day mortality (*p* = 0.032). In conclusion, we observed a high prevalence of sarcopenia among polytrauma patients, further increasing in older patients. In addition, sarcopenia was identified as a predictor for 30-day mortality, underlining the clinical significance of identification of low muscle mass on a CT scan that is already routinely obtained in most trauma patients.

## 1. Introduction

Sarcopenia, a disease characterized by progressive loss of skeletal muscle mass (SMM), muscle strength, and physical performance, is a common condition among senior adults [1]. It is associated with a wide range of adverse outcomes and leads to a decreased quality of life and increased mortality [1,2,3,4]. The clinical relevance of sarcopenia has been extensively described in various disorders [5,6,7,8]. In addition, the predictive value of skeletal muscle mass measurements for complications has been demonstrated in various areas of surgery, such as general, vascular, colorectal, and liver transplant surgery [7,9,10].

With worldwide increased life expectancy and associated increased incidence in geriatric polytrauma patients, prevalence of sarcopenia in trauma patients is expected to increase as well [11,12]. To date, prevalence of sarcopenia in polytrauma patients is largely unknown. Evidence on adverse outcome related to sarcopenia in polytrauma patients is scarce and quality of available studies is limited [13,14,15]. One limiting factor is the heterogeneity in polytrauma patients, including all age groups, different trauma mechanisms and a variety in pre-existing medical conditions. In addition, use of different definitions for sarcopenia hampers the comparison of studies in trauma patients [1,16,17]. Nevertheless, on arrival of a trauma patient, a total-body computed tomography (CT) scan is obtained almost routinely, and CT-based measurement of skeletal muscle mass is the gold standard for quantification of SMM. Therefore, the abdominal CT scan offers the opportunity to identify patients with low muscle mass directly.

The aim of this pilot study was to investigate the prevalence of low muscle mass in older polytrauma patients, as well as to explore the relation between low muscle mass and mortality, complications, and inflammatory response. We hypothesize that a low skeletal muscle mass on abdominal CT, as an indicator of sarcopenia, is common in polytrauma patients of increased age and that it could aggravate their clinical outcome by increasing the risk of complications. In addition, we hypothesize that low muscle mass is a predictor of 30-day or in hospital mortality and that it induces an increased early inflammatory response during hospitalization.

## 2. Patients and Methods

Approval for this study was acknowledged by the local ethics committee of the Maastricht University Medical Centre (MUMC+, Maastricht, the Netherlands). Requirement for informed consent was waived because of the retrospective nature of this study.

### 2.1. Patients and Determination of Muscle Mass

Data from polytrauma patients (injury severity score (ISS) ≥16) who were admitted to a single level-I trauma center between January 2015 and December 2019 were assessed for study eligibility. Patient records were retrospectively checked for presence of an abdominal CT scan and patient height. Only when both abdominal CT scan on admission and height were available, patient data were included for analysis. Measurement of SMM was performed using abdominal CT images by two independent observers using OsiriX Lite 11.0.2 open software on transverse slides of abdominal CT scan at the level of third lumbar (L3) vertebra as described before [18,19]. All measurements were performed in a semi-automated fashion by setting tissue of interest threshold at −30 to +110 Hounsfield Units (HU) for skeletal muscle [20]. Automatically generated areas of interest were corrected manually. Total muscle mass area was automatically calculated and displayed in square centimeters.

Skeletal mass index (SMI), a derivative of the skeletal muscle mass (SMM) [21], was then calculated using the following equation:Skeletal mass Index = Skeletal muscle mass (SMM)/height²(1)

To estimate prevalence of sarcopenia in the study population, sarcopenia was defined according to cutoff values for SMI as described by Prado et al. [18]. These values were determined at 52.4 cm²/m² and 38.5 cm²/m² for males and females, respectively. Two independent investigators measured the L3 muscle area of all patients and these data were used to calculate the interobserver agreement. Intra-observer agreement was assessed by repeating 50 L3-measurements 6 months after initial analysis.

### 2.2. Clinical Outcome

Complications were retrieved from patient data by two observers. For this purpose, complications (pneumonia, urinary tract infection, delirium and mortality (both in hospital and 30-day)) and duration of intensive care unit (ICU) admission and hospitalization were all scored. Diagnosis of pneumonia was based on chest radiographs and antibiotic treatment [22]. Urinary tract infection was defined as positive urinary culture and initiation of antibiotic therapy [23]. Delirium was diagnosed by a geriatrician in patients with altered mental status and if they received medical treatment [24]. In addition, inflammatory variables (leukocytes and C-reactive protein (CRP) on admission, after 24 and 48 h) were evaluated. Data from patients with severe head trauma who deceased within 24 h were included in the analysis of prevalence of sarcopenia. However, their data were excluded from analysis of complications.

### 2.3. Statistical Analysis

Frequencies are presented as absolute numbers and percentages. Continuous data is presented as mean (± standard error of the mean). Normal distribution was tested using Kolmogorov–Smirnov test. Differences between groups were analyzed using Pearson χ² test for dichotomous variables. Confidence intervals were calculated using logistic regression analysis. First, univariate analysis was performed to select parameters directly related to adverse outcome. Dependent variables that were identified in univariate analysis were subsequently entered into a multivariate logistic regression analysis. The influence of sarcopenia on 30-day mortality was determined using a log rank test.

The interobserver agreement (R.S., M.A.) of L3 muscle index assessment of sarcopenia was analyzed by the Pearson correlation index. Two-tailed P values less than 0.05 were considered significant. All statistical analyses were performed using SPSS (version 25.0; SPSS Inc. Chicago, IL, USA).

## 3. Results

### 3.1. Patients

Data from 846 polytrauma patients were assessed for eligibility. In 428 patients, no abdominal CT-scan was available. Further exclusion was due to missing data regarding patient height (*n* = 179) and death within 24 hours due to brain injury (*n* = 1). Therefore, data from 239 polytrauma patients were included in analysis of prevalence and from 238 for complications, excluding the single patient who died of severe brain injury within 24 hours. Patient demographics are listed in Table 1. One-hundred fifty nine of 239 (66.5%) patients were male, and 80 of 239 (33.5%) were female. Average age was 49 years (range 6–89, SD 21.45), with a non-significant distribution among gender (48 (± 20) and 53 (± 24) years for males and females, respectively, *p* = 0.078). For the total population, the following means were observed: ISS 26.7 (± 9.9), body mass index (BMI) 25.0 kg/m² (± 4.3), Charlson Comorbidity Index 1.7 (± 2.1). Patients were hospitalized for an average of 19 days (± 17). Mean L3 SMI for males and females was 57.4 cm²/m² (± 10.24) and 42.7 cm²/m² (± 7.82), respectively.

### 3.2. Prevalence

Prevalence of sarcopenia using the criteria as defined by Prado et al. was 80 out of 239 patients (33.5%), of whom 52 (65%) were male and 28 out of 80 (35%) were female (see Table 2). Mean SMI in the sarcopenic group was 42.8 (± 6.9) while this was 57.5 (± 10.5) for the non-sarcopenic group. SMI in males was higher compared to females (57.4 cm²/m² (± 10.2) vs 42.7 cm²/m² (± 7.8), respectively). In the older cohort (> 80 years), prevalence increased to 85%, with an even distribution between males and females (83.3% and 85.7%, respectively). Patients defined as sarcopenic were older than non-sarcopenic patients (57 years (± 22.9) and 46 years (± 19.5), respectively, *p* < 0.0001). ISS was comparable in sarcopenic (ISS = 25.9) and non-sarcopenic (ISS = 27.2) patients. The relation between age, ISS, and SMI is represented in Figure 1.

### 3.3. In Hospital Mortality

Eighteen (7.6%) patients died within one month or during hospital admission. Log rank analysis identified sarcopenia as an independent predictor of 30-day mortality (*p* = 0.032, Figure 1). In univariate analysis, sarcopenia (*p* = 0.045) was identified as a predictor for 30-day or in-hospital mortality. In addition, age (*p* = 0.005), Charlson Comorbidity Index (*p* = 0.001), ISS (*p* = 0.026), surgical procedures (*p* = 0.038), and hospital length of stay (*p* = 0.004) were significant predictors of mortality within one month. In the binary logistic regression analysis age (OR, 1.08; 95% CI 1.01–1.165; *p* = 0.018), ISS (OR, 1.19; 95% CI 1.04–1.20; *p* = 0.003) and hospital length of stay (OR, 0.83; 95% CI 0.74–0.92; *p* < 0.0001) remained as independent predictors of mortality within one month or during hospital admission. Logistic regression results are summarized in Table 3.

### 3.4. Complications and Inflammatory Response

The incidence of complications was 68 (45.3%) in non-sarcopenic patients and 41 (52.6%) in sarcopenic patients (*p* = 0.28). No significant differences were observed between sarcopenic and non-sarcopenic cohorts regarding prevalence of pneumonia (*p* = 0.25), urinary tract infection (*p* = 0.34), delirium (*p* = 0.085) and ICU (*p* = 0.48) or hospital length of stay (*p* = 0.29). The inflammatory response in sarcopenic patients showed a significant increase in leukocyte levels at 48 hours compared to non-sarcopenic patients (11.95 (± 3.64) vs 10.08 (± 3.04), respectively, *p* = 0.002). The other time points (admission and 24 hours after admission) showed no difference for leukocyte levels (*p* = 0.18 and *p* = 0.45, respectively).

### 3.5. Interobserver Agreement of CT Based Muscle Measurement by Osirix

Interobserver agreement analysis of all measurements showed a strong and significant correlation (R² = 0.99; *p* < 0.0001). The interclass correlation coefficient (ICC) of sarcopenia assessment by CT image analysis using Osirix was 0.99 (*p* < 0.0001) with a Cronbach alpha of 0.99. The interobserver coefficient of variation (CV) was 10.1%. Intraobserver agreement analysis of 50 L3-measurements showed a significant correlation (0.863; *p* < 0.0001) in the repeated measurements. 

## 4. Discussion

The clinical significance of sarcopenia, defined in this study as a decreased skeletal muscle mass (SMM), is becoming evident in many fields of medicine. The depletion of muscle mass is a risk factor for infection during hospitalization in both non-cancer patients and cancer patients [9,25,26]. Furthermore, it has been revealed that sarcopenia in cancer patients is associated with treatment toxicity, poor functional status, increased length of hospital stay, prolonged rehabilitation care, and increased mortality (Huaiying 2019) [9,18,27,28,29]. The prevalence of sarcopenia in community-dwelling populations is up to 29%, ranging from 12% to 60% in patients with colorectal cancer [30,31]. However, despite the direct clinical relevance, the prevalence of sarcopenia in polytrauma patients is unclear. This study identified a prevalence of 34% in the overall polytrauma study group, increasing to more than 80% in polytrauma patients aged over 80 years. Furthermore, our data indicate a relationship between sarcopenia and 30-day or in-hospital mortality, regardless of gender, underlining the clinical significance of sarcopenia in polytrauma patients.

Our data are in line with other studies that have described the prevalence of sarcopenia in various populations [32,33,34]. In general populations, von Haehling and colleagues describe a prevalence of 5–13% in people aged 60–70 years, increasing to 11–50% in those aged 80 or above. These numbers are comparable to the prevalence of sarcopenia in populations suffering from obstructive pulmonary disease (COPD), renal failure and cancer [5,35,36]. The present study shows a higher prevalence of sarcopenia in polytrauma patients compared to other studies using the same age groups. Although the reason for this higher prevalence is beyond the scope of the current analysis, it raises questions on the causal relationship between sarcopenia and accidents. Sarcopenia is associated with an increased risk of falling [37], and in a part of the current population, a fall was registered as mechanism of trauma. Another aspect that has to be taken into account in interpretation of the high prevalence in our population is the applied definition of sarcopenia. In the present study, the criteria for skeletal muscle mass as described by Prado were used. Other definitions of sarcopenia use functional tests in addition to the quantification of muscle mass, such as the EWGSOP-II definition that uses hand-grip strength and gait speed [1]. However, in the polytrauma population these parameters are not readily obtainable. This limitation is mostly due to the extremity injuries frequently present in polytrauma patients, but also admission to the ICU and (prolonged) ventilation makes it impossible to obtain these data. Obtaining data during the admission is not straight forward, as loss of muscle mass and muscle function occurs within days during bedrest [38]. In contrast, a total-body CT scan is performed almost routinely upon arrival of a polytrauma patient, providing detailed imaging of the abdomen. The quantification of skeletal muscle mass is therefore readily available, and our study indicates that this parameter is directly related to survival.

The relation between sarcopenia and mortality, as indicated by many publications, appears in the univariate analysis of our data as well. In addition, a clear prediction of 30-day mortality was found. The identification of sarcopenia in these analyses is in line with available literature, showing increased mortality within one year of diagnosing sarcopenia [39,40]. A recent study by Leeper et al. indicates sarcopenia to be a strong predictor of 6-month post-discharge mortality in elderly trauma patients [41]. However, in the multivariate analysis in our study, we were unable to maintain sarcopenia as an independent risk factor for 30-day or in-hospital mortality. This is likely due to the small sample size in our study, as well as the stronger correlation between mortality and other factors such as age and trauma severity. Notwithstanding this effect, in the study by Leeper et al., an association is described between mortality and a variety of factors including ICU and hospital length of stay and injury scoring systems like Injury Severity Score and Abbreviated Injury Scale. Our data are in accordance with their findings.

Obtaining an abdominal CT scan is becoming a more and more widespread routine in trauma patients [42]. Our data showed the same trend, as the percentage of patients excluded for missing CT data decreased per year. Surprisingly, data from routinely obtained trauma CT scans are only seldom used for muscle mass measurement in clinical practice. We believe that the addition of skeletal mass measurement analysis in trauma patients has a direct beneficial effect, since treatment of low muscle mass can be initiated immediately, and further loss of muscle mass could be prevented. One consideration in using only CT derived data is, however, the mentioned cutoff values for the L3 index, which are used to estimate sarcopenia. As these values are based on obese patients with cancer, caution is advised when translating these values to other populations. Ideally, cutoff values for sarcopenia should be established within each specific patient population and BMI category. A much larger sample size would be required to undertake cutoff values analyses by gender and by Body Mass Index (BMI). It is important to realize that there has been mounting evidence on the clinical importance of BMI in the field of traumatology. An observational prospective study of Childs et al. shows an increased risk of infections, acute renal failure, length of ICU stay, hospital length of stay, and duration of mechanical ventilation in polytrauma patients with a BMI > 30 [43]. In addition, our analyses suggest differences in prevalence of sarcopenia between different BMI ranges, with an increasing portion of sarcopenic trauma patients in higher BMI ranges. However, it is important to acknowledge that BMI does not distinguish between muscle and fat tissues. Increase in routinely obtained CT-scanning gives the opportunity to assess CT-based anthropometric parameters of fat in addition to muscle mass. A recent study of Poros et al. showed that CT-based assessment of abdominal fat is suitable in revealing pathologic body composition in trauma patients [44]. Abdominal fat measurements might add valuable information in relation to complications and mortality in trauma patients in future studies.

The current study was a pilot study, where a retrospective analysis of collected data was performed as a first step to elucidate the clinical impact of sarcopenia in trauma patients. This study has therefore limitations that have to be kept in mind in interpretation of the results. One of the limitations of this study is its retrospective nature. Because only data from patients with an abdominal CT were included in our analysis, there is a potential inclusion bias. Still, all cases were retrieved from a regional prospective trauma registry, thereby limiting the effect of potential selection bias. Another limitation is the heterogeneity of the study population. Traumatic events occur in any age group with different grades of severity. In order to limit the effect of heterogeneity, only patients with an ISS ≥ 16 were included, and age-specific analysis was obtained regarding the prevalence of sarcopenia and its complications. Finally, only CT data were analyzed, and other markers of frailty such as functional and nutritional status were not included in our analysis.

In future studies, functional tests should, if possible, be included in the criteria for sarcopenia. Moreover, the cut-off values for skeletal mass index on CT measurements may require different cut-off points based upon reported outcome in this specific population.

## 5. Conclusions

The prevalence of sarcopenia in elderly polytrauma patients is high, reaching 85% in people over 80 years old. Our analysis also shows that sarcopenia is an independent predictor for 30-day mortality. Since abdominal CT scans are now almost routinely obtained in trauma patients, we advocate the measurement of skeletal mass measurement to detect decreased muscle mass early. Early identification of people at risk for sarcopenia will lead to early diagnosis and therapy, and more awareness will help to prevent additional loss of muscle mass during the admission after trauma.

## Figures and Tables

**Figure 1 jcm-09-02046-f001:**
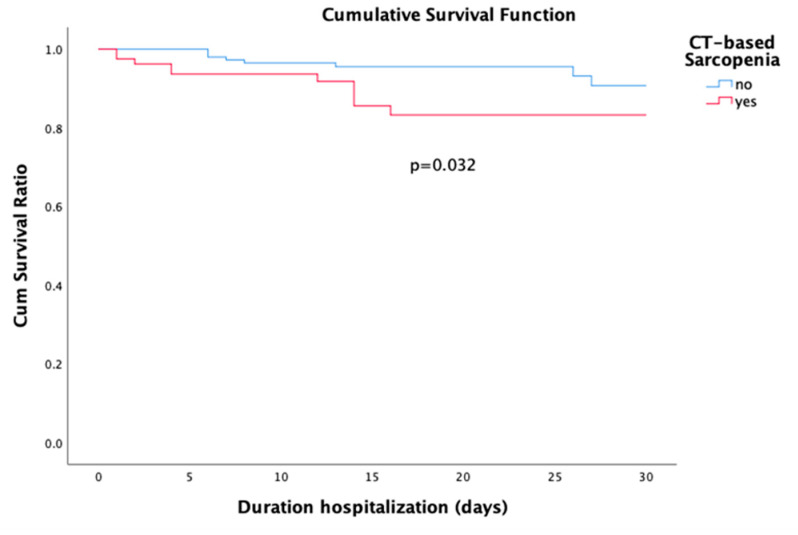
Survival curves for sarcopenia and no sarcopenia.

**Table 1 jcm-09-02046-t001:** Patient characteristics.

	No. of Patients (%)	Mean (SD)	Sarcopenic	Non-Sarcopenic	Significance
Gender					
Male	159 (66.5%)		51	108	
Female	80 (33.5%)		30	50	
Age					
Male		48 (± 20)	52 (± 21.9)	46 (± 18.7)	*p* = 0.046
female		53 (± 24)	66 (± 22.7)	46 (± 21.0)	*p* < 0.0001
>80	20 (8.3%)				
Male		84 (± 3)	5	1	
Female		83 (± 2)	12	2	
BMI					
<18.5	7 (2.9%)		4 (1.7%)	3 (5.1%)	*p* = 0.174
18.5–24.9	120 (50.5%)		54 (22.7%)	66 (27.7%)	*p <* 0.0001
25–29.9	88 (36.9%)		20 (8.4%)	68 (28.6%)	*p* =0.006
>30	23 (9.7%)		2 (0.8%)	21 (8.8%)	*p* = 0.005
Length of hospital stay (days)		19 (± 17)	17.6 (± 15.9)	20.1 (± 17.3)	*p* = 0.29
Length of stay ICU (days)		6 (± 8)	5.4 (± 7.5)	6.2 (± 8.6)	*p* = 0.48
Injury severity score		26.7 (± 9.9)	25.9 (± 9.3)	27.1 (± 10.2)	*p* = 0.34
Charlson comorbidity index					
0–1	145 (60.9%)		37	108	
>2	94 (39.9%)		44	50	
SMI					
Male <52.4	52 (21.8%)				
Female <38.5	30 (11.8%)				
Complications					
Pneumonia	43 (18.1%)		17 (20.9%)	26 (16.4%)	
Urinary tract infection	11 (5.5%)		2 (2.5%)	9 (5.6%)	
Delirium	51 (21.0%)		22 (27.2%)	29 (18.4%)	
Mortality within 1 month	18 (7.5%)		10 (12.3%)	8 (5.1%)	*p* = 0.032
Mortality within 1 year	8 (3.4%)		4 (4.9%)	4 (2.5%)	
Mortality after 1 year	5 (2.1%)		3 (3.7%)	2 (1.3%)	
Number of patients requiring emergency (<24 h) surgery	108 (45.4%)		33 (41.3%)	75 (47.5%)	
Number of patients requiring ICU admission	181 (76.1%)		127 (80.4%)	54 (67.5%)	

BMI: body mass index; SD: standard deviation; ICU: intensive care unit; SMI: skeletal mass index.

**Table 2 jcm-09-02046-t002:** Sarcopenia prevalence in polytrauma population.

	General Population (%)	Age 60–79 (%)	Age ≥ 80 (%)
Group			
Females	11.8 (*n* = 28/239)	13.6 (*n* = 9/66)	60 (*n* = 12/20)
Males	21.8 (*n* = 52/239)	24.2 (*n* = 16/66)	25 (*n* = 5/20)

**Table 3 jcm-09-02046-t003:** Logistic regression analysis of mortality within one month.

		Univariate Analysis		Multivariate Analysis	
	Mortality	Odds Ratio	*p*-Value	Odds Ratio	*p*-Value
Gender					
Male	9/150	1			
Female	9/79	2.14 (0.82–5.6)	0.115		
Age		1.05 (1.02–1.08)	0.005		
BMI		0.93 (0.82-1.04)	0.21		
Sarcopenia					
No	8/157	1			
Yes	10/81	2.62 (0.99–6.93)	0.45		
Charlson Comobidity Index		1.46 (1.20–1.77)	0.001		
Injury Severity Score		1.05 (1.00–1.09)	0.026	1.19 (1.00–1.41)	0.05
Surgery during hospitalization					
No		1			
Yes		0.36 (0.14–0.98)	0.38		
Inflammatory parameters					
Plasma CRP at hospitalization		0.97 (0.87–1.07)	0.55		
Plasma Leukocyte at hospitalization		1.01 (0.98–1.03)	0.59		
Plasma CRP after 24 h		0.99 (0.99–1.01)	0.85		
Plasma Leukocytes after 24 h		1.08 (0.95–1.24)	0.25		
Plasma CRP after 48 h		0.99 (0.99–1.00)	0.19		
Plasma leukocytes after 48 h		1.13 (0.94–1.36)	0.20	1.77 (1.06–2.96)	0.029
Complications					
Pneumonia					
No	14/195	1			
Yes	4/43	1.33 (0.41–4.25)	0.63		
Urinary tract infection					
No	18/227				
Yes	0/11		0.33		
Delirium					
No	11/187	1			
Yes	7/51	2.55 (0.93–6.94)	0.06		
ICU length of stay		0.99 (0.93–1.06)	0.84		
Hospital length of stay		0.89 (0.82–9.63)	0.004	0.67 (0.50–0.89)	0.006

CRP: C-reactive protein.

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
