# Peer review of "High Prevalence of Sarcopenia in Older Trauma Patients: A Pilot Study"

_jcm, 2020, doi:10.3390/jcm9072046_

Round 1

Reviewer 1 Report

The authors provided a very well written manuscript that assessed the prevalence of sarcopenia in elderly polytrauma patients. The two most important limitations that I have found in the paper (cut-off values that considered other populations and the absence of force measurements) were already nicely addressed by the authors in the paper. Although the retrospective nature of the study can be considered a limitation, the paper also adds important findings to the scientific literature. The authors should be commended for presenting a clear and transparent interpretation of their results. I’m looking forward to reading future studies including strength and functional tests in this population.

Just a couple of things to consider:

  1. Consider other works in the field: https://pubmed.ncbi.nlm.nih.gov/31745608/
  2. Did the authors consider measuring abdominal fat as well? This could be valuable information for future studies.
  3. “Sarcopenia, a disease characterized by progressive loss of skeletal muscle mass (SMM), muscle strength, and physical performance, is a common condition among senior adults [1].” The study did not measure muscle strength (the main parameter that defines sarcopenia, EWGSOP2). The authors should consider adding grip strength and sit-and-stand tests into clinical practices. There are also validated questionnaires that assess muscle strength. The inclusion of these tests into clinical practices will strengthen future studies.
  4. Intra-observer agreement data should be reported in the manuscript. I recommend the authors to include some other important measurements of reliability, such as CV and TEM. These data should be, ideally, presented in a table.
  5. Is it possible to show the number and % of participants with different BMIs in Table 1? The number of participants with BMI from 18.5 to 24.9 (healthy weight), from 25 to 29.9 (overweight), and > 30 (obesity)? I’m also wondering if the study has enough power to run a statistical analysis comparing sarcopenia between the different BMI groups.
  6. Did the authors assess sex or gender? If it was sex, please change the nomenclature. Also, please use either the terms males and females or women and men. For gender, males and females would be more appropriate.

Minor:

Lines 19: Please add the age range here.

Lines 20 – 22: Please add statistical analysis results here.

Lines 22: Please correct the typo “sarcopenie”.

Line 24: There an extra space here.

Lines 48 – 52: Please state the aim of the study before the hypothesis. Is there any hypothesis for gender-related differences? Please include a hypothesis for each of the outcomes (mortality, inflammatory response, gender, etc.).

Lines 83 – 84: Please add references here.

Lines 103 – 107: Is it possible to provide a flow chart that represents the included and excluded participants?

Lines 158: Can you please explore the complications that can be induced by sarcopenia?

Lines 161 – 162: Please remove the space between the numbers and the %.

Lines 160 – 162: I’d add here that these results were observed regardless of gender (or sex if gender was not in fact assessed).

Author Response

Consider other works in the field: https://pubmed.ncbi.nlm.nih.gov/31745608/

Re: We thank the reviewer for the suggestion. We added a paragraph in the discussion section about the clinical relevance of BMI in trauma patients and the use of CT-based anthropomorphic measurements, based on the reference as suggested by the reviewer.

Did the authors consider measuring abdominal fat as well? This could be valuable information for future studies.

Re: Although we agree that this information could be valuable for additional studies, we have adhered to the current guidelines in measuring muscle mass, including the previously used thresholds in the setting of the software. The raw scans are still in our database and the suggested analysis could be performed in the future.

However, we added a literature overview emphasizing the clinical relevance of BMI in the trauma population and the future role of CT-based abdominal fat measurements in the discussion (line 236- 242).

“Sarcopenia, a disease characterized by progressive loss of skeletal muscle mass (SMM), muscle strength, and physical performance, is a common condition among senior adults [1].” The study did not measure muscle strength (the main parameter that defines sarcopenia, EWGSOP2). The authors should consider adding grip strength and sit-and-stand tests into clinical practices. There are also validated questionnaires that assess muscle strength. The inclusion of these tests into clinical practices will strengthen future studies.

Re: Thank you for this valuable addition. As we have indicated in the discussion (lines 208-209), the functional tests are not available in trauma patients due to, i.e. fractures of the extremities. The paragraph in the discussion has been extended in order to further clarify our limitations in obtaining functional data.

Intra-observer agreement data should be reported in the manuscript. I recommend the authors to include some other important measurements of reliability, such as CV and TEM. These data should be, ideally, presented in a table.

Re: We reported the inter- and intra-observer agreement data in the manuscript (line 172). We agree with the reviewer to include other measurements of reliability. Therefore, we calculated the inter-observer coefficient of variation (line 175)

Is it possible to show the number and % of participants with different BMIs in Table 1? The number of participants with BMI from 18.5 to 24.9 (healthy weight), from 25 to 29.9 (overweight), and > 30 (obesity)? I’m also wondering if the study has enough power to run a statistical analysis comparing sarcopenia between the different BMI groups.

Re: As suggested by the reviewer, we have made changes to table 1, showing the numbers and percentages of patients within different BMI ranges. Interestingly, these data suggest that sarcopenia prevalence decreases with higher BMI, as documented. These data were not part of our primary or secondary research questions, and therefore our sample size was not sufficient to run statistical analyses on these subgroups. We placed a comment on these results in the discussion by emphasizing the role of BMI in trauma patients and by adding the following sentence to this new paragraph: “In addition, our analyses suggest differences in prevalence of sarcopenia between different BMI ranges, with an increasing portion of sarcopenic trauma patients in higher BMI ranges.”

Did the authors assess sex or gender? If it was sex, please change the nomenclature. Also, please use either the terms males and females or women and men. For gender, males and females would be more appropriate.

Re: We thank the reviewer for pointing this out. In this study, we assessed gender and adjusted the nomenclature in the entire manuscript to “gender” and “males and females”.

Lines 19: Please add the age range here.

Re: We thank the reviewer for the suggestion. We added “range 60-79 years” to the document.

Lines 20 – 22: Please add statistical analysis results here.

Re: We added the p-values of the statistical analysis as suggested by the reviewer.

Lines 22: Please correct the typo “sarcopenie”.

Re: We corrected “sarcopenie” to “sarcopenia”.

Line 24: There an extra space here.

Re: We deleted the extra space between “as” and “a”.

Lines 48 – 52: Please state the aim of the study before the hypothesis. Is there any hypothesis for gender-related differences? Please include a hypothesis for each of the outcomes (mortality, inflammatory response, gender, etc.).

Re: Thank you for your remark. As suggested by the reviewer we reorganized the paragraph stating the aim before the hypothesis. Furthermore, we added the hypothesis on 30-day and in hospital mortality as well as on inflammatory response. Gender-related differences were considered beyond the scope of this study and were not added to the main hypotheses.

Lines 83 – 84: Please add references here.

Re: We have inserted references regarding the used definitions for pneumonia, urinary tract infection and delirium.

Lines 103 – 107: Is it possible to provide a flow chart that represents the included and excluded participants?

Yes, that will follow asap. Given the short response time as requested by the editorial office this will follow in the next days.

Lines 158: Can you please explore the complications that can be induced by sarcopenia?

Re: As suggested by the reviewer, we elaborated on the risks induced by depletion of muscle mass in more detail. We added complication details concerning infections in non-cancer and cancer patients and added complications as treatment toxicity, poor functional status, increased length of hospital stay, prolonged rehabilitation and increased mortality as extensively described in mainly oncological literature.

Lines 161 – 162: Please remove the space between the numbers and the %.

Re: We deleted the space between “34” and “%”.

Lines 160 – 162: I’d add here that these results were observed regardless of gender (or sex if gender was not in fact assessed).

Re: As suggested by the reviewer, we added “regardless of gender” in line 161.

Reviewer 2 Report

interesting retrospective analysis using an innovative methodology that has been used far too rarely to date. Just a small hint - perhaps it makes sense to remove children and adolescents from the evaluation, since no sarcopenia is to be expected here and the incision may be falsified?
Could the mortality, complications and length of stay be calculated again for the age cohorts 60-79 and >80? However, the cohorts may be too small to achieve significant results.

Author Response

(The authors gave the same response as above.)
